# An Evolving Multivariate Time Series Compression Algorithm for IoT Applications

**DOI:** 10.3390/s24227273

**Published:** 2024-11-14

**Authors:** Hagi Costa, Marianne Silva, Ignacio Sánchez-Gendriz, Carlos M. D. Viegas, Ivanovitch Silva

**Affiliations:** 1UFRN-PPgEEC, Postgraduate Program in Electrical and Computer Engineering, Federal University of Rio Grande do Norte, Natal 59078-970, Brazil; hagi.costa.062@ufrn.edu.br (H.C.); marianne.silva@penedo.ufal.br (M.S.); 2Campus Arapiraca, Federal University of Alagoas, Penedo 57200-000, Brazil; 3UFRN, Department of Computing Engineering and Automation, Federal University of Rio Grande do Norte, Natal 59078-970, Brazil; ignaciogendriz@dca.ufrn.br (I.S.-G.); carlos.viegas@ufrn.br (C.M.D.V.)

**Keywords:** IoT, TinyML, evolving algorithm, data compression, OBD-II edge

## Abstract

The Internet of Things (IoT) is transforming how devices interact and share data, especially in areas like vehicle monitoring. However, transmitting large volumes of real-time data can result in high latency and substantial energy consumption. In this context, Tiny Machine Learning (TinyML) emerges as a promising solution, enabling the execution of machine-learning models on resource-constrained embedded devices. This paper aims to develop two online multivariate compression approaches specifically designed for TinyML, utilizing the Typicality and Eccentricity Data Analytics (TEDA) framework. The proposed approaches are based on data eccentricity and do not require predefined mathematical models or assumptions about data distribution, thereby optimizing compression performance. The methodology involves applying the approaches to a case study using the OBD-II Freematics ONE+ dataset, which is focused on vehicle monitoring. Results indicate that both proposed approaches, whether parallel or sequential compression, show significant improvements in execution time and compression errors. These findings highlight the approach’s potential to enhance the performance of embedded IoT systems, thereby improving the efficiency and sustainability of vehicular applications.

## 1. Introduction

The Internet of Things (IoT) has significantly transformed the operation of industries by enabling real-time communication and data exchange among interconnected devices [1]. This paradigm shift has driven advances in sectors such as healthcare, manufacturing, and transportation, increasing operational efficiency, enabling predictive maintenance, and driving innovation in products and services [2]. Projections indicate that the number of connected devices could exceed 75 billion by 2025, underscoring the importance of the IoT in the future of industry [3].

However, the enormous volume of data generated by IoT devices presents considerable challenges in terms of storage and processing [4]. IoT networks, particularly those handling multivariate time series data from multiple sensors, require efficient methods to manage this large amount of information [5]. Traditional storage systems often struggle to scale to handle such large and rapid data streams, which raises concerns about their sustainability [6]. Furthermore, processing this data in real-time to derive actionable insights requires computational resources and advanced algorithms, which can be expensive and complex to implement [3,7].

These challenges are particularly evident in the context of autonomous and/or connected vehicles, where IoT plays an important role [8]. Vehicles equipped with multiple sensors generate extensive data streams, encompassing information on speed, engine performance, environmental conditions, and driver behavior [9]. Managing and analyzing this data in real-time is essential to ensure safety and efficiency of driving systems. For instance, real-time data processing enables the detection of potential road hazards, optimizes route planning, and provides immediate feedback to vehicle control systems, demonstrating the practical application of IoT in the automotive industry [10]. Additionally, beyond vehicle monitoring, applications such as industrial machinery tracking, smart grid data analysis, and even environmental monitoring are examples where similar challenges of multivariate data compression and processing on resource-limited devices are important.

Considering the volume and complexity of IoT data, particularly in time series form, it is essential to use data compression techniques to minimize storage and processing costs [11]. Univariate techniques, such as the Tiny Anomaly Compressor (TAC) [12], have already proven effective in reducing data volume while preserving information by identifying anomalies in time series from a single sensor.

In this context, TinyML emerges as a promising approach, enabling the deployment of machine-learning algorithms on low-power devices, such as sensors and microcontrollers [13]. This approach is relevant for embedded systems and IoT, where energy efficiency and local processing are important [14]. Compared to conventional ML techniques, TinyML offers lower energy consumption and reduced latency, which is important in IoT applications where power resources are constrained and connectivity may be intermittent. Additionally, TinyML models allow data processing closer to the source, which minimizes the need for constant data transmission, thus saving energy and improving efficiency [15].

The Tiny Anomaly Compressor (TAC) is an example of a TinyML technique designed to operate on resource-constrained devices. Based on the Typicality and Eccentricity Data Analysis (TEDA) framework, TAC is an approach that, unlike traditional methods that rely on data density, evaluates whether a sample is an outlier by examining its typicality and eccentricity [12]. Eccentricity measures the extent to which a data point deviates from the others in the dataset, while typicality assesses how representative or common that point is within the same dataset [16,17]. Implementing TAC on TinyML devices is advantageous because it allows the execution of compression and anomaly-detection algorithms directly on the sensors, taking advantage of local processing to minimize the need for transmitting large volumes of data. The main research problem addressed in this study is the limitation of univariate techniques, like TAC, in effectively handling multivariate data from multiple interconnected sensors, where variables may interact and exhibit complex dependencies that are essential for accurate anomaly detection.

Thus, to address this limitation, this work proposes two extensions of TAC to a multivariate framework: the Multivariate Parallel Tiny Anomaly Compressor (MPTAC) and the Multivariate Sequential Tiny Anomaly Compressor (MSTAC). These approaches allow TAC to be applied in contexts where multiple variables are monitored simultaneously, accounting for correlations and interactions among them. A case study using vehicle data collected by a Freematics ONE+ OBD-II device was conducted in order to compare the performance of MPTAC and MSTAC. In summary, this work presents significant innovations by expanding the capabilities of TAC for multivariate IoT applications, thus addressing both data volume and resource constraints.

**Extension of TAC:** Development of the MPTAC and the MSTAC for multivariate data compression and anomaly detection, which enables efficient handling of interrelated variables.**Processing efficiency:** Improvement in real-time analysis by considering correlations between variables and facilitating on-device data compression.**Cost and energy efficiency:** Reduction in energy consumption and data transmission requirements by leveraging TinyML processing capabilities, demonstrating advantages over traditional ML approaches in constrained IoT environments.**Practical validation:** Effectiveness demonstrated through a case study using vehicle data, showcasing potential for broader IoT scenarios.

Preliminary results indicate that both approaches offer significant improvements in execution time and compression, with MPTAC particularly enhancing data processing efficiency for embedded IoT systems. While this study focuses on vehicular data, the methodology has potential applications across diverse IoT domains, including smart cities, industrial automation, and environmental monitoring.

The remainder of this paper is organized as follows. Section 2 presents related works that influenced our defined methodology and implementation. Section 3 provides the essential background information needed to understand the problem and the context of our algorithms. Section 4 details the proposed algorithms. Section 5 presents the case study by outlining the research questions and providing a detailed description of the dataset and the metrics used for evaluating the algorithms. Section 6 discusses the main results, and finally, Section 7 presents conclusions and promising directions for future research.

## 2. Related Works

The processing and compression of temporal data have been the focus of extensive research due to the exponential growth in data generated by IoT devices, monitoring systems, and other sources [11]. Several techniques have been developed to address the challenges associated with the efficient storage and analysis of these data [18]. This section reviews the main approaches to time series compression.

In [19], the authors present a compression approach for resource-constrained IoT applications that handle univariate and multivariate time series data. The method integrates the lifting scheme implementation of the Discrete Wavelet Transform (DWT) with an error-bound lossy compressor known as Squeeze (SZ). This combination reduces data size while maintaining essential data quality by denoising and smoothing the input signal for more effective compression. The algorithm has been applied to various IoT datasets, demonstrating its adaptability to different sensor data types and activities. However, the approach’s effectiveness depends on the stationarity and noise characteristics of the time series data, which can affect compression performance in multivariate cases where signal noise and variations across dimensions differ.

In the work [20], the authors introduce a compression algorithm tailored for non-equidistant multivariate integer time series data, specifically applied in the context of vehicle sensor data. The proposed algorithm, named Binary Shift Compression (BiSCo), operates by first adjusting signal values using a bit-shift method based on a defined maximum absolute error and a signal-specific factor. This is followed by a zero-order prediction model that compresses the data by storing only significant changes in the sensor readings. The algorithm is specifically designed for handling large volumes of sensor data generated by vehicle networks. Its real-time efficiency and high compression ratio make it suitable for automotive applications, but its design for integer-only data introduces limitations in contexts that require floating-point precision or highly dynamic data. This may be especially challenging in applications that need to maintain accuracy for signal variability in multivariate setups.

In the study [21], the authors propose a transfer-learning-based multimodal convolutional denoising autoencoder (M-CDAE) designed for compressing biosignals such as electrocardiogram (ECG), electromyogram (EMG), and electroencephalogram (EEG), which are commonly used for diagnosing chronic diseases. The proposed algorithm works by jointly compressing these signals into a unified representation before transmission. This approach reduces computational costs and enhances the battery life of wearable devices. The compression process is optimized by using transfer learning, which leverages pre-trained weights from similar datasets to improve the reconstruction quality of the biosignals from their compressed forms. This method, though effective for biosignals, could struggle with the unique characteristics of multivariate sensor data in IoT, where diverse signals may not correlate closely, affecting transfer learning’s generalization capability. Additionally, while transfer learning reduces training times, initial pre-training remains computationally demanding.

In the research presented in [22], the authors propose LFZip, a lossy compression algorithm for multivariate floating-point time series data. The algorithm employs a prediction-quantization-entropy coding framework, starting with the prediction of the next value in the time series using models such as Normalized Least Mean Square (NLMS) or neural networks. The prediction error is then quantized and the resulting data are entropy coded. LFZip has been applied to various time series datasets from domains like activity recognition, power consumption, and sensor data. The algorithm is designed to work with a user-specified maximum absolute error. Nonetheless, LFZip’s dependency on prediction models may reduce efficiency in non-stationary environments, particularly if changes in time series patterns are not quickly adapted by the neural network. This limitation is more pronounced in resource-constrained environments where computational overheads hinder real-time performance.

In [23], the authors propose a physiological signal-compression algorithm tailored for mobile health (mHealth) applications, leveraging an optimized Spindle Convolutional Autoencoder (SCAE). This method is designed to compress multimodal biosignals such as ECG, EEG, and EMG by encoding the signals into a lower-dimensional space before reconstruction. Its use of channel pruning and quantization enables high-efficiency compression, but its focus on biosignals, which may not generalize well to IoT sensor data due to inter-signal dependency characteristics in health data. Consequently, this algorithm might require adaptation to effectively handle more loosely related multivariate IoT data.

In their research [24], the authors propose a near-lossless compression method for time series data, specifically designed to optimize data transmission and storage in resource-constrained environments such as IoT devices. The algorithm operates in two main stages: a data transformation phase followed by entropy encoding. In the data transformation phase, the method uses quantization to reduce the precision of floating-point numbers, followed by statistical analysis and deviation coding to minimize data redundancy. The resulting data are then encoded using an entropy-based compression algorithm, such as adaptive arithmetic coding, to further reduce the data size. The algorithm has been applied to various datasets, including those from wearable sensors and household power consumption monitoring, making it suitable for a wide range of IoT applications. Despite its effectiveness, limitations arise with non-stationary multivariate data, where statistical assumptions may not hold, and quantization may compromise the integrity of individual signals.

In the article [25], the authors propose a sliding window-based time series compression algorithm named Window Delta (WD). This algorithm is designed to handle time series data characterized by frequent fluctuations, which can pose challenges for traditional compression methods. WD operates by applying a sliding window of size 3 to smooth the data, reducing the impact of fluctuations on the compression process. The algorithm computes intermediate values, termed w-deltas, which represent the difference among data points within the window and their average. While effective for data with rapid changes, WD’s approach may be constrained by its simplistic structure, which lacks the adaptability needed for multivariate data, especially in IoT settings where fluctuations are irregular and may vary in amplitude across dimensions.

In the investigation [26], the authors present a lossless compression algorithm called Ant, specifically designed for floating-point time series data commonly generated in IoT applications. The algorithm converts double-precision floating-point numbers into integer form, calculates the delta among consecutive values, and then applies Zigzag encoding to convert the delta into unsigned integers. This approach effectively minimizes the storage requirements by focusing on the significant bits of the encoded values while discarding unnecessary leading zeros. Ant is applied to various time series datasets, particularly those involving industrial monitoring and sensor data, demonstrating its utility in IoT scenarios where efficient storage and transmission of large data volumes are critical. Although Ant’s method for minimizing storage requirements by transforming floating-point values into integers is innovative, its applicability is limited in data with large variations between adjacent values, where precision is important. The conversion to integers can result in lossiness for multivariate datasets requiring high decimal precision.

From another perspective, ref. [12] proposes TAC, an anomaly-detection-based compression approach for time series. TAC compresses only the detected anomalies, making it efficient for data where normal patterns dominate, yet univariate TAC limitations underscore the necessity for advancements in multivariate monitoring scenarios. Expanding TAC to a multivariate framework can bridge this gap, especially in TinyML settings that require robust, resource-efficient anomaly compression. The main advantage of TAC is its ability to efficiently identify and represent anomalous events, which are considered critical information for compression.

Overall, while the reviewed techniques address various compression challenges, the need for scalable, multivariate-aware solutions for TinyML applications remains unfulfilled. By enhancing the multivariate capacity of TAC through MPTAC and MSTAC, our approach aims to address these limitations, particularly in adaptability and computational efficiency, which are essential for resource-limited devices.

Table 1 presents a comparative summary of these approaches, highlighting their key features and limitations. The table details whether the algorithm handles multivariate time series maintains floating-point precision, employs online learning, uses machine learning, and performs compression in a single stage. This comparative analysis emphasizes that while several methods attempt to tackle multivariate and resource-constrained challenges, none fully integrate these capabilities in a unified solution.

## 3. Theoretical Background

This section reviews the foundational concepts and existing methodologies that underpin our work, with a particular focus on the TEDA framework and the TAC.

### 3.1. Typicality and Eccentricity Data Analysis

The TEDA framework, derived from recursive density estimation algorithms, uses the concepts of typicality and eccentricity to distinguish normal from anomalous data samples [16]. Unlike traditional methods that rely on data density, TEDA assesses whether a sample is an outlier by examining its typicality and eccentricity. These measures are computed without requiring specific parameters or thresholds.

TEDA is designed to function without many of the restrictive assumptions that traditional statistical methods require. It does not assume independence among data samples, nor does it require a large sample size or prior knowledge of the data distribution, such as normality. This framework relies on data proximity and mutual distribution, allowing for flexible, non-parametric anomaly detection that adapts to diverse data types. TEDA’s threshold is selected to enable sensitivity to deviations, dynamically adapting based on data variability. This flexibility allows TEDA to effectively identify anomalies without capturing minor fluctuations, ensuring accurate and adaptive performance in real-time applications.

In TEDA, typicality reflects how well a data sample aligns with the general pattern of the dataset. This measure assesses the representativeness of each data point in relation to the dataset context, where a high typicality value indicates that the data point is consistent with established data trends. In contrast, a low typicality value implies that the data point deviates from these patterns, which may suggest anomalous behavior.

Eccentricity in TEDA measures the extent to which a data point diverges from the dataset’s central tendency. It is calculated based on the distance from the sample to the mean and variance in the data up to that point, quantifying deviation without fixed parameters or thresholds. Data points with high eccentricity are identified as significantly divergent. When paired with low typicality, high eccentricity signals an outlier, equipping TEDA with the ability to detect anomalies in dynamic, evolving datasets.

The eccentricity of a sample, ξ, for Euclidean distance, can be represented as follows [27]:(1)$$\xi_{k}\left(x_{k}\right)=\frac{1}{k}+\frac{{\left(\mu_{k}-x_{k}\right)}^{T}\left(\mu_{k}-x_{k}\right)}{k\sigma_{k}^{2}},\;\sigma_{k}^{2}>0,\;k>1$$
where k∈N is the sampling instant, $x_{k}\in R^{n}$ is the *k*-th *n*-dimensional sample (n∈N), $\mu_{k}\in R^{n}$ is the recursively updated mean (Equation (Equation 2)), and $\sigma_{k}^{2}\in R$ is the recursively updated variance (Equation (Equation 3)).
(2)$$\mu_{k}\left(x_{k}\right)=\left\{\begin{array}{cc} 0, & if\;k=0\\ x_{k}, & if\;k=1\\ \frac{k-1}{k}\mu_{k-1}+\frac{x_{k}}{k}, & if\;k>1 \end{array}\right.$$
(3)$$\sigma_{k}^{2}\left(x_{k}\right)=\left\{\begin{array}{cc} 0, & if\;k\leq 1\\ \frac{k-1}{k}\sigma_{k-1}^{2}+\frac{\left|\right|x_{k}-\mu_{k}{\left|\right|}^{2}}{k-1}, & if\;k>1 \end{array}\right.$$

The typicality of a given sample $x_{k}$ at the *k*-th iteration can be described as the complement to eccentricity, as follows [27]:(4)$$\tau_{k}\left(x_{k}\right)=1-\xi_{k}\left(x_{k}\right)$$

Additionally, ref. [27] defines the normalized eccentricity, which can be computed as follows:(5)$$\zeta_{k}\left(x_{k}\right)=\frac{\xi_{k}\left(x_{k}\right)}{2},\;\sum_{i=1}^{k}\xi_{k}\left(x_{k}\right)=1,\;k>1$$

To distinguish normal state data from abnormal state data, it is essential to establish a comparison threshold. For anomaly detection, the mσ threshold is commonly used [28]. However, this approach requires an assumption about the distributional characteristics of the analyzed data, such as assuming a Gaussian distribution [27]. Chebyshev’s inequality, on the other hand, can be applied to any data distribution. It asserts that the probability of data samples deviating more than mσ from the mean is at most $1/m^{2}$, where σ represents the standard data deviation [29].

The condition that yields results equivalent to Chebyshev’s inequality, without making any assumptions about the data or its independence, can be expressed as follows:(6)$$\zeta_{k}>\frac{m^{2}+1}{2k}$$
where *m* is the comparison threshold [27].

Finally, Algorithm 1 presents the details of TEDA in pseudocode form.   
**Algorithm 1:** TEDA
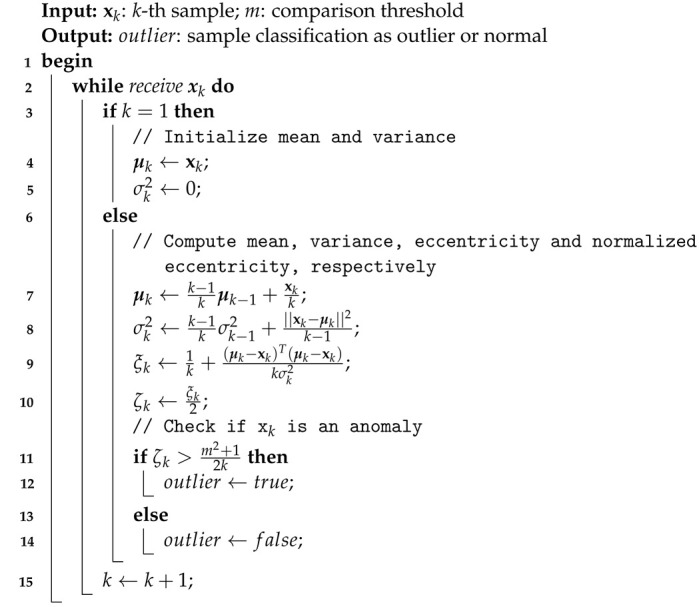


### 3.2. Tiny Anomaly Compressor (TAC)

The TAC algorithm builds upon the principles of typicality and eccentricity discussed in TEDA [27] to compress single-variable time series data using anomaly detection. Like TEDA, TAC does not require predefined mathematical models or assumptions about data distribution. It leverages recursive calculations, ensuring efficiency with low computational cost, minimal memory usage, and reduced processing power demands [12]. This approach is effective for time series data that exhibit spatial or temporal correlation, such as sensor-generated data streams, by focusing specifically on anomalies within that single dimension.

TAC introduces differences compared to the TEDA framework outlined in Algorithm 1. The first difference in TAC is the use of a dynamic “anomaly window”. This window tracks the number of anomalies detected since the last sample was saved. In addition, a new hyperparameter, windowLimit, is introduced to specify the number of anomalies required to consider the window full. This hyperparameter controls the model’s sensitivity to concept drifts: higher values mean that more anomalies are needed before the model resets and saves a new sample, thus tuning the response to changes in the signal.

When the anomaly window is not full, no new samples are saved. Once the window reaches its capacity and an additional anomaly is detected, that sample is preserved. This indicates a potential concept drift in the time series. The window then resets, and the model’s internal parameters—such as *k*, mean, and variance—are reset to their initial states to start a new window. Finally, the pseudocode for TAC method is provided in Algorithm 2.
**Algorithm 2:** Tiny Anomaly Compressor (TAC)
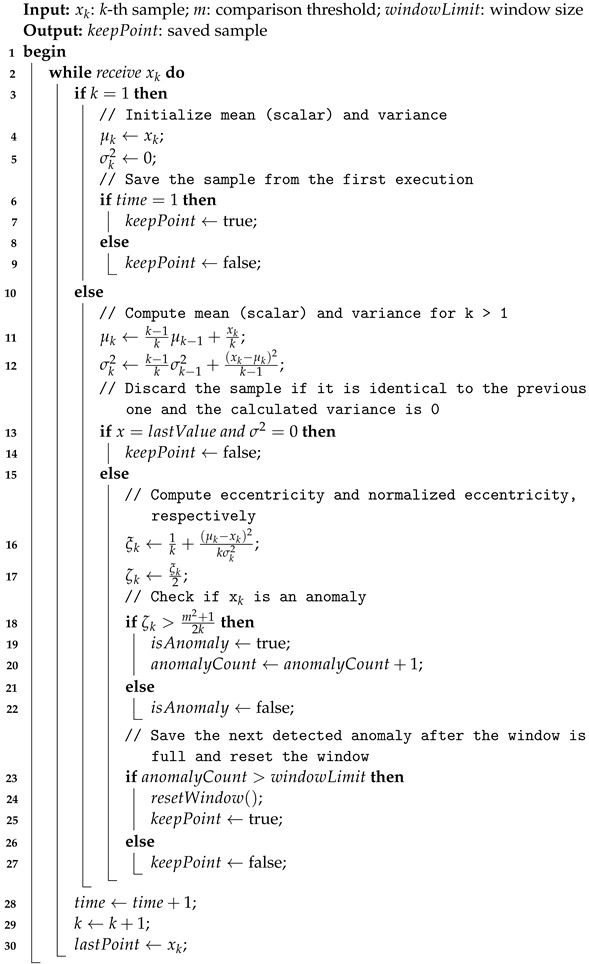


## 4. Proposed Algorithm

In this section, we present the development of the algorithms underpinning MPTAC and MSTAC. Both algorithms are designed to compress multivariate data, but they differ in their approaches to handling multiple variables.

### 4.1. Multivariate Parallel Tiny Anomaly Compressor

The proposed work extends the TAC algorithm from its original single-variable application to a multidimensional framework, enhancing its applicability in scenarios where multiple interrelated variables are monitored simultaneously. This extension is depicted in Figure 1, highlighting how TAC’s functionality is enhanced for managing and analyzing multidimensional data.

However, we adapt the anomaly detection process to account for correlations and interactions among multiple variables. This adaptation enables the algorithm to identify anomalies not only within individual dimensions but also in their relationships.

The resulting algorithm, named MPTAC, builds on the foundational logic of the TAC. While the original TAC was implemented and evaluated using simplified forms of Equations (Equation 1)–(Equation 5) for univariate time series [12], this study generalizes the approach to handle the case where $x_{k}$ is an *n*-dimensional sample, applying the full version of the TEDA framework.

The primary difference between TAC and MPTAC lies in the treatment of input data. TAC processes a single variable $x_{k}$ at a time, calculating the mean $\mu_{k}$ and variance $\sigma_{k}^{2}$ as scalar quantities. In contrast, MPTAC processes vectors of multiple variables $x_{k}\in R^{n}$, computing the mean $\mu_{k}$ as a vectorial quantity (variance $\sigma_{k}^{2}$ remains a real value). This approach enables MPTAC to capture correlations among variables, which are important for detecting anomalies in a multivariate context.

Moreover, MPTAC incorporates the calculation of the vectorial distance ${\left(\mu_{k}-x_{k}\right)}^{T}\left(\mu_{k}-x_{k}\right)$, allowing the algorithm to identify deviations not only within individual dimensions, but also across the overall vector. This capability makes MPTAC more effective in detecting complex anomalies that would be overlooked by univariate methods. Finally, the pseudocode for this parallel compression approach is provided in Algorithm 3.
**Algorithm 3:** Multivariate Parallel Tiny Anomaly Compressor
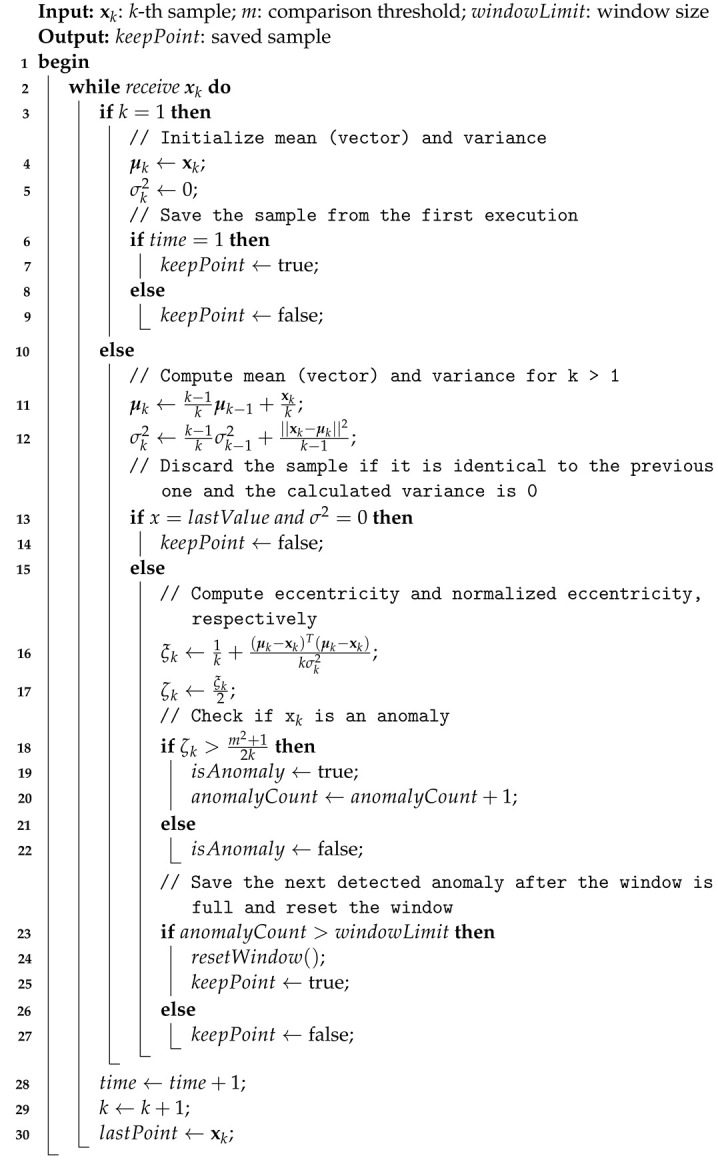


### 4.2. Multivariate Sequential Tiny Anomaly Compressor

A sequential multivariate compressor, named the MSTAC, is an approach that operates by applying the unidimensional TAC algorithm independently to each variable in a multivariate dataset. Specifically, the MSTAC algorithm is composed of *n* separate instances of the TAC algorithm, where *n* corresponds to the number of variables in the dataset.

Each instance of the TAC algorithm within the MSTAC framework is dedicated to compressing a single variable. This means that for each variable, the algorithm performs sequential compression tailored to that variable’s unique statistical properties and characteristics. By handling each variable independently, MSTAC ensures that the compression process is optimized for the specific dynamics and variations inherent in each data stream.

The corresponding flowchart of the MSTAC algorithm is illustrated in Figure 2, which provides a visual representation of the sequential compression process applied to each variable.

The pseudocode for this sequential compression approach is provided in Algorithm 4. In this algorithm, $x_{k}\left[v\right]$ represents the value of the *v*-th variable in the *k*-th sample, m[v] denotes the corresponding *m* value used by the TAC algorithm for the *v*-th variable, and windowLimit[v] specifies the window size used to control the compression process for the *v*-th variable. The MSTAC algorithm thus performs compression by processing each variable in sequence (for instance, $x_{k}\left[3\right]$ is processed only after $x_{k}\left[2\right]$ has been processed), with each instance of TAC operating independently to compress its designated variable.
**Algorithm 4:** Multivariate Sequential Tiny Anomaly Compressor
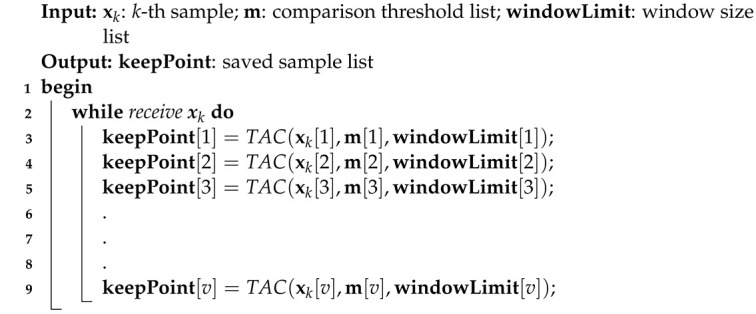


Finally, Figure 3 illustrates the processing flow of the TAC, MPTAC, and MSTAC algorithms, highlighting their key differences. The TAC processes univariate data, computing mean and variance to detect anomalies in a single dimension. The MPTAC extends this concept to multiple dimensions by processing multivariate data in parallel, with vectorized calculations of mean and variance, enabling simultaneous anomaly detection across several variables.

The MSTAC, on the other hand, follows a sequential approach, applying the TAC independently to each dimension of the multivariate data. This approach allows for more granular anomaly detection, with decisions made separately for each variable. Both MTAC and MSTAC are built upon the TAC foundation, but they offer different methods for handling the complexity of multivariate data.

### 4.3. Algorithm Characteristics of MPTAC and MSTAC

The computational complexity of the MPTAC and MSTAC algorithms is O(n), where *n* represents the number of monitored variables, meaning that data dimensionality is the sole factor impacting performance. Both algorithms are designed for continuous data streams and scale linearly; however, MPTAC stands out in practical efficiency due to parallel processing, which reduces execution time, especially in high-dimensional scenarios such as IoT applications and multivariate systems.

The flexibility of these algorithms allows tuning of anomaly-detection sensitivity through two main parameters: anomaly window size and *m*-threshold. The anomaly window size defines the number of consecutive anomalies required before a data point is retained. A smaller window size makes the algorithm more responsive, capturing frequent or sporadic anomalies and useful for detecting subtle deviations, although it may lead to false positives in more volatile data. A larger window size, on the other hand, requires more consistent anomalies before retention, offering robustness in highly variable data streams.

The *m*-threshold defines the eccentricity level required to classify a point as an anomaly. Lower *m* values make the algorithm more sensitive to minor variations, detecting smaller deviations but possibly increasing false positives. Higher *m* values are more suitable for data with natural variability, as they decrease sensitivity to minor fluctuations, though they may miss subtler anomalies.

Additionally, the algorithms differ in handling correlations between variables, impacting both compression performance and anomaly detection. While MPTAC calculates a single eccentricity score for the multivariate sample, capturing collective patterns among correlated variables, MSTAC treats each variable independently, increasing precision in detecting deviations specific to each dimension. In resource-constrained applications, such as IoT, selecting representative variables from strongly correlated groups can reduce redundant processing by monitoring only those with the highest informational value.

Finally, these algorithms are robust in handling edge cases in data, such as very smooth or highly erratic data streams, through appropriate parameter adjustments. For smoother data, lowering the *m*-threshold and anomaly window size increases sensitivity to subtle anomalies, preserving data integrity during compression. In volatile time series, raising these parameters prevents excessive data retention, ensuring that only significant deviations are preserved.

These characteristics make MPTAC and MSTAC versatile and efficient algorithms, adapting to different data types and operational conditions to maximize compression without compromising anomaly-detection accuracy.

## 5. Case Study

The objective of this case study is to assess the performance of the proposed MPTAC algorithm in terms of compression errors, processing efficiency, and data compression effectiveness. This involves comparing MPTAC with various instances of the TAC algorithm, each configured with different parameter settings, as well as with the MSTAC. In both cases, linear interpolation was the technique used for decompression.

### 5.1. Research Questions

This section addresses the following research questions:1.How efficient is the MPTAC algorithm in terms of compression ratio compared to the MSTAC algorithm across various parameter settings?2.How does the MPTAC algorithm compare to MSTAC in terms of precision, measured by *root mean squared error (RMSE)*, when evaluated with different parameters and when adjusted to the same parameterization?3.How does the processing time of the MPTAC algorithm compare to the processing time of the MSTAC algorithm for data compression?4.What are the main advantages and disadvantages of the MPTAC algorithm compared to the MSTAC algorithm, considering reconstruction error metrics?

### 5.2. Database Selection

This study considered the dataset provided by the Conect2AI (https://conect2ai.dca.ufrn.br/, accessed on 7 September 2024) group. It contains 10.365 records from automotive sensors, sampled at a rate of 1 Hz (1-s sampling interval), using the Freematics ONE+ device, which is an Arduino-compatible vehicle telematics prototyping platform [30]. It includes a total of 36 columns, each representing a different feature of the vehicle. For this analysis, we specifically focused on five variables: battery voltage, engine load, RPM, speed, and throttle—since they represent a mix of dynamic data, which varies rapidly, and more stable data, as noted in Table 2. Furthermore, we note that no preprocessing steps were applied to the dataset, as the raw data was utilized directly.

In addition, as shown in Figure 4, the data exhibit significant variability in the RPM and speed, while battery voltage remains relatively stable. This selection enables the performance evaluation of compression algorithms across both types of data, ensuring a comprehensive assessment of their effectiveness.

### 5.3. Evaluation Metrics

To evaluate the performance of a compression encoder for time series data, three key characteristics are considered: compression ratio, processing speed, and compression error [31].

**Compression ratio**—This metric measures the effectiveness of a compression technique and is defined as follows:(7)$$\rho =\frac{s^{\prime }}{s}$$
where $s^{\prime }$ is the size of the compressed representation and *s* is the size of the original time series. Other key metrics related to the compression ratio include the *compression rate (CR)* and the *compression factor (CF)*:(8)CR=1−ρ
(9)$$CF=\frac{1}{\rho }$$

**Processing speed**—The following are the definitions of the two metrics associated with processing speed:*Compression time (CT)*: Compression time is the amount of time required to convert the original data into its compressed form. It is a critical metric for evaluating the efficiency of a compression algorithm, especially in applications that require real-time or near-real-time data processing;*Decompression time (DT)*: Decompression time is the amount of time required to reconstruct the original data from its compressed form. This metric is crucial for applications where quick access to the original data is necessary after compression.

**Compression error**—Compression error measures the fidelity of the reconstructed time series relative to the original. Various metrics can be used to assess this fidelity, like *mean squared error (MSE)*, *root mean squared error (RMSE)*, *signal-to-noise ratio (SNR)*, *normalized cross correlation (NCC)* and *compression $F_{\beta }$ coefficient (C$F_{\beta }$)* [12]. These metrics allow not only the evaluation of reconstruction accuracy but also insight into the impact of decompression errors on specific applications.
(10)$$MSE=\frac{1}{n}\sum_{i=1}^{n}{\left(x_{i}-{\hat{x}}_{i}\right)}^{2}$$
(11)$$RMSE=\sqrt{MSE}$$
(12)$$SNR=\frac{\sum_{i=1}^{n}\frac{x_{i}^{2}}{n}}{MSE}$$
(13)$$NCC=\frac{1}{n}\sum_{i=1}^{n}\frac{\left(x_{i}-\mu_{x}\right)\left({\hat{x}}_{i}-\mu_{\hat{x}}\right)}{\sigma_{x}*\sigma_{\hat{x}}}$$
(14)$$CF_{\beta }=\left(1+\beta^{2}\right)\frac{CR*NCC}{\left(\beta^{2}*CR+NCC\right)}$$
where $x_{i}$, ${\hat{x}}_{i}$, $x_{peak}$, $\mu_{x}$, $\mu_{\hat{x}}$, $\sigma_{x}$ and $\sigma_{\hat{x}}$ are, respectively, the original data points, the reconstructed data points, the maximum value in the original time series, the original data points mean, the reconstructed data points mean, the original data points standard deviation and the reconstructed data points standard deviation. β is a positive real parameter used to adjust the relative importance of each metric in the $CF_{\beta }$ calculation. The parameter is selected so that the NCC is β times more important than the CR. Finally, when β=1, the equation assigns equal weight to both metrics, effectively becoming a simple harmonic mean.

In critical applications, such as real-time monitoring and anomaly-detection systems, errors can result in significant distortions in the signal pattern, affecting decision-making accuracy. For instance, a high *MSE* or low *SNR* may indicate a substantial deviation between the original and decompressed data, which could hinder the ability to detect trends or anomalies. Similarly, a low *NCC* value could mean that important variations are not preserved, impacting predictive maintenance applications that rely on high fidelity to original data trends.

The *C$F_{\beta }$* metric further supports balancing compression ratio and fidelity, adjusting the weight of each based on application demands. For applications where fidelity is critical, a higher β emphasizes the *NCC* over the *CR*, ensuring that the decompressed data maintains close correlation with the original signal. Thus, these metrics collectively guide the trade-off between compression efficiency and error tolerance, aligning the decompression quality with application-specific requirements.

## 6. Results

In this section, we present the results obtained from the evaluation of the MPTAC and MSTAC algorithms in the proposed case study.

To ensure optimal performance of the MPTAC and MSTAC algorithms, a grid search was conducted for each algorithm. This process involved systematically exploring various combinations of the key parameters windowLimit (ranging from 2 to 30) and *m* (ranging from 0.1 to 2.1). These values were chosen based on [12]. The goal of this optimization was to identify the parameter settings that maximize the algorithms effectiveness.

The evaluation metric used in this study was the average $CF_{\beta }$, with β=1, which assigns equal importance to both the CR and the NCC. This balanced approach ensures that the algorithms achieve high compression efficiency while maintaining the integrity of the original data.

The results are presented separately for each algorithm in the following subsections, facilitating a more in-depth analysis of the performance of MPTAC and MSTAC.

### 6.1. MPTAC—Parallel Compression

Following the grid search optimization, the hyperparameters for which MPTAC demonstrated the best performance were found to be windowLimit = 5 and m=0.2. These settings resulted in an average ${\bar{CF}}_{\beta }=0.8170$, with a compression rate of 86.25% (and a compression factor of 7.27). Table 3 presents these values alongside additional compression error metrics.

Figure 5 illustrates the performance of the algorithm, showing the original values (blue line), the saved values (red dots), and the decompressed values (red line) for each variable within the interval between samples 4100 and 4600, selected for improved data visualization.

Supplementary to this, Figure 6 displays the probability density functions (PDFs) for each variable, comparing the distributions of the original and decompressed data. These figures, particularly Figure 6, validate the effectiveness of the MPTAC algorithm, demonstrating its capability to maintain data integrity while achieving significant compression.

### 6.2. MSTAC—Sequential Compression

A grid search was conducted for each instance of MSTAC to identify the optimal parameters for the sequential approach. The best-performing parameters are summarized in Table 4, which also presents the corresponding compression metrics. The table presents the variables analyzed, along with their respective values for the window limit and compression factor *m*. It includes key performance indicators such as CR, CF, RMSE, NCC, and ${\mathrm{CF}}_{\beta }$ to evaluate the compression performance.

For instance, the speed variable achieves the highest compression rate of 96.83%, with a compression factor of 31.60, while maintaining a strong correlation (NCC = 0.8552) and relatively low RMSE (13.60). Conversely, variables such as RPM shows higher RMSE values, suggesting greater reconstruction error. These results indicate that the performance of MSTAC varies across different variables, depending on their specific characteristics and the optimal window size determined through the grid search.

Figure 7 shows the original values (blue line), the saved values (red dots), and the decompressed values (red line) for each variable within the interval between samples 4100 and 4600.

Figure 8 displays the probability density functions (PDFs) for each variable, comparing the distributions of the original and decompressed data. Similar to MPTAC, these results, particularly Figure 8, validate the effectiveness of the MSTAC algorithm by demonstrating its ability to maintain data integrity while achieving significant compression performance.

In addition, to compare the algorithms, Table 5 consolidates the results from Table 3 and Table 4. Generally, the variables show similar hyperparameters and compression metrics across both approaches. However, there is a notable exception for the speed variable: with windowLimit = 27 in the MSTAC approach, the compression factor is 31.60, which is more than four times higher than that of the MPTAC, without a significant negative impact on the compression error metrics.

Despite this, the MPTAC algorithm demonstrated significantly faster performance, processing a sample 4.17 times faster than the MSTAC. Consequently, for the dataset considered, MPTAC proves to be a more suitable choice, effectively balancing compression efficiency with processing speed.

Finally, we compare the performance of MPTAC against MSTAC using the same defined parameters (windowLimit = 4 and m=0.1). These values were selected by averaging the parameter values from the previous section, excluding the parameters of the TAC instance in MSTAC applied to the speed variable. This exclusion was necessary due to the significant discrepancy in parameter values for speed compared to the other variables, as shown in Table 6. This comparison highlights the performance differences between the two algorithms under consistent settings.

In Figure 9, we present a visual representation to facilitate the interpretation of the results. Figure 9 contains two subfigures: (a) it compares the overall compression metrics by variable and algorithm, and (b) it presents the comparison of the absolute root mean square error (RMSE) between the variables and algorithms, highlighting differences in accuracy between MPTAC and MSTAC for the same parameterization conditions.

### 6.3. Discussion of Results

In this subsection, we discuss the effectiveness of the MPTAC and MSTAC algorithms based on the obtained results and address the proposed research questions.

Evaluating the effectiveness of the MPTAC algorithm in terms of CR (Question 1), MPTAC and MSTAC, it is observed that MPTAC, with the parameters windowLimit = 5 and m=0.2, presented a CR of 86.25%, while MSTAC achieved a varied compression ratio, with the best rate of 96.83% for the speed variable when windowLimit = 27 and m=0.2. Despite the higher compression rate of MSTAC for the speed variable, MPTAC showed a competitive compression rate in other variables, such as battery voltage and engine load. These results suggest that MPTAC is effective in maintaining a high compression ratio while preserving data integrity.

When comparing the MPTAC algorithm to MSTAC in terms of precision, as measured by RMSE (Question 2), it is evident that the performance varies with different parameter settings—Table 5, the MPTAC and MSTAC algorithms show distinct performance characteristics in terms of RMSE. For battery voltage, MPTAC with parameters (0.2, 5) achieves an RMSE of 0.56, compared to MSTAC with parameters (0.1, 4), which yields an RMSE of 0.54, indicating slightly better precision for MSTAC. For engine load, MPTAC with parameters (0.2, 5) has an RMSE of 5.33, while MSTAC with parameters (0.1, 5) results in an RMSE of 5.43, showing MPTAC’s advantage in this case. In the RPM category, MPTAC (0.2, 5) has an RMSE of 295.73, whereas MSTAC (0.1, 5) yields 294.68, demonstrating a marginally better precision for MSTAC. For speed, MPTAC with parameters (0.2, 5) has an RMSE of 11.89, compared to MSTAC with parameters (0.2, 27), which has an RMSE of 13.60, indicating better performance by MPTAC. Finally, for throttle, MPTAC (0.2, 5) shows an RMSE of 5.52, whereas MSTAC (0.1, 3) has an RMSE of 5.00, highlighting MSTAC’s superior precision. These observations suggest that the choice between MPTAC and MSTAC may depend on the specific parameters used and the variable being considered, as each algorithm offers distinct advantages in different contexts.

When evaluated with the same parameterization (m=0.1, windowLimit=4), Table 6, the MPTAC and MSTAC algorithms exhibit similar performance in terms of RMSE, with minor variations across specific variables. MSTAC demonstrated a slight advantage in precision for battery voltage (0.54 vs. 0.55), RPM (289.20 vs. 291.00), and speed (11.93 vs. 11.72), while MPTAC performed better for engine load (5.31 vs. 5.49) and throttle (5.23 vs. 5.56). These differences are subtle, indicating that both algorithms are comparably effective, with the choice between them depending on the relative importance of precision for specific variables within the application’s context.

Regarding the processing time comparison between MPTAC and MSTAC (Question 3), processing time is an important factor in selecting a compression algorithm, especially in applications where efficiency and speed are essential. In the case of the MPTAC and MSTAC algorithms, a significant difference in processing time was observed. MPTAC demonstrated noticeably faster performance, with an average processing time per sample of 0.1997 ms, which is approximately 4.17 times faster than MSTAC, which has an average time of 0.8326 ms per sample. The superior efficiency of MPTAC is especially advantageous in scenarios that require low-latency real-time processing, where the speed of data compression is as crucial as its efficiency. The advantage in processing time makes MPTAC a preferred choice for applications that require high speed, such as embedded systems in vehicles or IoT devices, where fast processing can directly impact overall performance and user experience. On the other hand, although MSTAC presents higher processing time, it can still be considered in situations where compression efficiency is prioritized over speed.

Complementary to these findings, the evaluation of the MPTAC and MSTAC algorithms revealed different trade-offs between compression and accuracy, aspects that are fundamental in applications with bandwidth and processing constraints. In situations where compression rate is a priority, such as remote sensor networks, a more aggressive compression is advantageous to reduce the volume of transmitted data. However, this choice can introduce small losses in accuracy, which in some contexts, such as environmental monitoring, is acceptable to preserve bandwidth and storage efficiency. On the other hand, in applications that require high accuracy, such as real-time anomaly detection in automotive systems, a more moderate compression rate is preferred, which ensures a more accurate reconstruction of the data and, therefore, greater reliability in the analysis.

In addition, the performance of the MPTAC and MSTAC algorithms strongly depends on the adequate calibration of parameters such as the size of the analysis window and the thresholds for compression and anomaly detection. During the study, a careful adjustment of these parameters was performed through empirical tests on the automotive dataset, allowing a balance to be reached between compression efficiency and reconstruction accuracy. This calibration process is also adaptable to other domains, such as health monitoring and smart grids, where specific data characteristics require adjustments to optimize algorithm performance. Thus, the choice between algorithms and their parameters depends both on the specific application requirements and on the conditions and limitations of the available data.

Finally, the main advantages and disadvantages of the MPTAC and MSTAC algorithms are summarized in Table 7. This comparison highlights the key strengths and limitations of each approach, focusing on important factors such as processing time, compression ratio, and data integrity. By examining these aspects, the table provides a clear overview of the trade-offs between the two algorithms.

In summary, the choice between MPTAC and MSTAC depends on the specific application priorities. If processing speed is the primary concern, MPTAC is more advantageous. However, if the compression ratio is more critical, MSTAC may be preferable, especially for variables where it demonstrates a superior compression ratio. These findings extend beyond the automotive context; they can be applied to various IoT applications, such as wearable devices and smart cities, where efficiency in data transmission and storage is essential.

### 6.4. Limitations of MPTAC and MSTAC

While the MPTAC and MSTAC algorithms offer frameworks for multivariate time series compression, they also exhibit certain limitations.

Scalability concerns—As the number of variables increases, both MPTAC and MSTAC maintain a computational complexity of O(n) relative to the number of variables. However, the constant multiplicative factor for MSTAC is higher due to its sequential processing approach, which results in longer processing times. While MPTAC’s parallel handling of multidimensional data can lead to increased memory usage and processing time, its recursive and online processing nature allows it to maintain efficient performance even with large sample sizes. Significant performance constraints for MPTAC would likely only arise in extreme scenarios involving a high number of variables combined with very limited computational power. In contrast, the sequential processing of MSTAC may lead to greater inefficiencies in real-time applications as the number of variables grows.

Performance in highly dynamic environments—Both algorithms face challenges in highly dynamic environments where concept drift occurs. Although the anomaly window mechanism in MPTAC and MSTAC provides some level of adaptability, this mechanism may not be sufficient in rapidly changing data distributions. In these situations, reinitialization of the algorithms may be required to maintain accuracy. This could disrupt continuous monitoring and anomaly detection in scenarios where real-time response is critical, such as in industrial IoT or smart grid systems.

Parameter sensitivity—The effectiveness of both MPTAC and MSTAC rely heavily on careful tuning of hyperparameters, such as the comparison threshold and window size. Incorrect selection of these parameters can significantly impact performance, potentially resulting in a higher rate of false positives or missed anomalies. Scenarios that most challenge parameter tuning are those with highly erratic (highly dynamic) time series or, conversely, extremely smooth series, where inappropriate parameter choices can reduce detection accuracy. In diverse applications, these parameters may need continuous adjustment, which could limit the generalizability of the algorithms across varying use cases without prior fine-tuning.

Handling of strong correlations—In MSTAC, the approach of processing variables independently makes it less susceptible to misclassifying one variable due to “double weighting” from strongly correlated variables, as each variable is evaluated on its own. However, this independence assumption may lead to missed anomalies that arise from interdependencies among variables, limiting its effectiveness in capturing the full complexity of data in multivariate systems. MPTAC, by processing variables in parallel as a collective sample, can better handle interdependent variables but may be more sensitive to strongly correlated variables. In such cases, these correlations might carry greater influence when classifying a sample as an anomaly.

Handling outliers—Both MPTAC and MSTAC rely on outliers as indicators of context shifts, using them to identify and save relevant samples. The control of hyperparameters, such as *m* and the anomalywindowsize, helps ensure that only significant outliers are considered, avoiding the influence of less relevant anomalies. The eccentricity-based anomaly-detection mechanism in both algorithms further supports this by flagging outliers before compression, allowing for tailored handling. Given this, handling outliers is not a weakness of the algorithms; rather, it is an integral feature that enhances their accuracy in capturing meaningful data shifts and improves the quality of compressed data in dynamic environments.

## 7. Conclusions

This study aimed to evaluate and compare the performance of the proposed and implemented compression algorithms MPTAC and MSTAC for time series data compression. The analysis was performed using an automotive dataset, focusing on compression ratio metrics, data fidelity (compression error metrics), and processing speed.

The results showed that MPTAC outperformed MSTAC on several key metrics. Specifically, MPTAC demonstrated significantly lower RMSE for most variables analyzed, indicating data compression with less loss of accuracy. This result is relevant for applications that require high fidelity of data after compression.

In terms of processing speed, MPTAC demonstrated remarkably superior performance, processing samples 4.17 times faster than MSTAC. This efficiency is important for applications where rapid compression and decompression are required, such as embedded and real-time systems.

In addition, MPTAC consistently maintained data integrity, as evidenced by the NCC, which remained high even with the most aggressive compression. This suggests that MPTAC can effectively balance compression efficiency with preserving data quality, which is essential for analysis and decision making based on compressed data.

For future research, it is recommended to explore the application of MPTAC and MSTAC across various datasets and contexts to evaluate their versatility and robustness. Investigating the adaptability of these algorithms to different window sizes and parameters could provide further insights into their performance and applicability. Additionally, conducting analyses on the application of the algorithms in real-time environments and on hardware platforms such as the Freematics ONE+ will enhance our understanding of their practical utility. These directions not only underscore the potential impact of this research but also inspire further investigations into the development of data processing solutions.

## Figures and Tables

**Figure 1 sensors-24-07273-f001:**
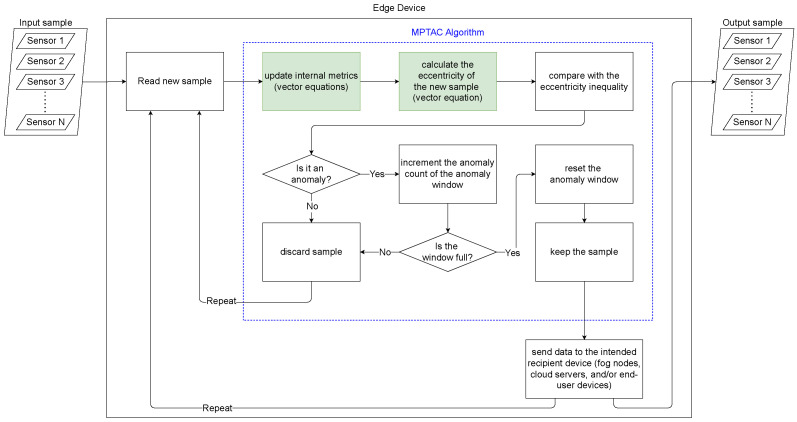
Flowchart of the MPTAC algorithm.

**Figure 2 sensors-24-07273-f002:**
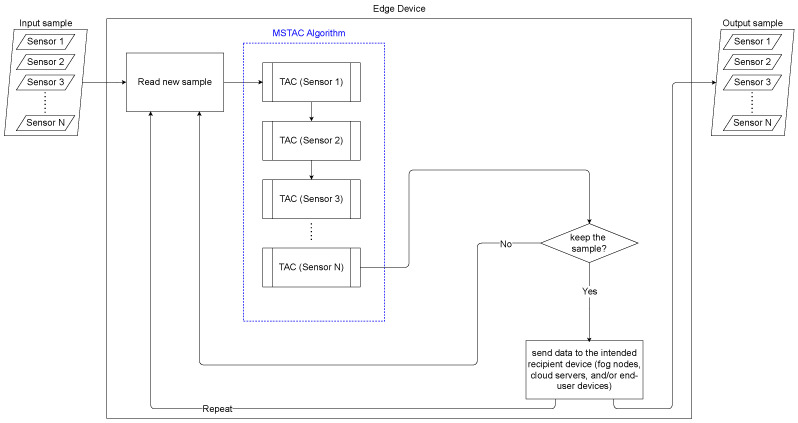
Flowchart of the MSTAC algorithm.

**Figure 3 sensors-24-07273-f003:**
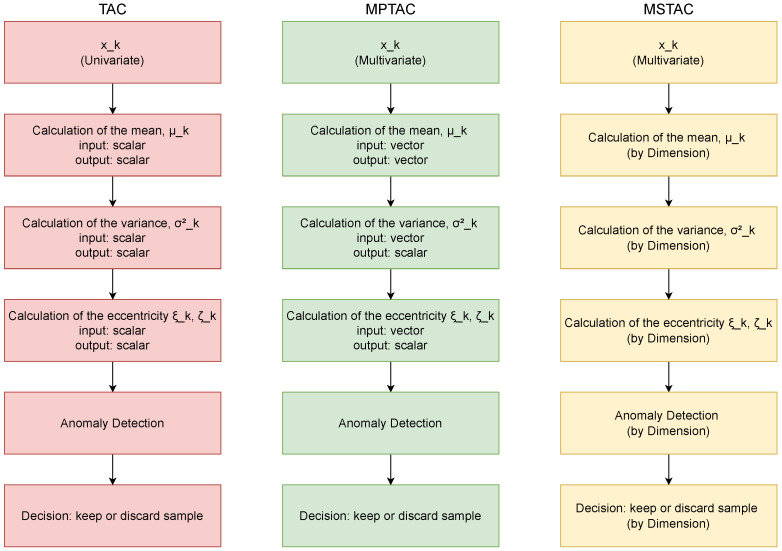
Differences between TAC, MPTAC, and MSTAC.

**Figure 4 sensors-24-07273-f004:**
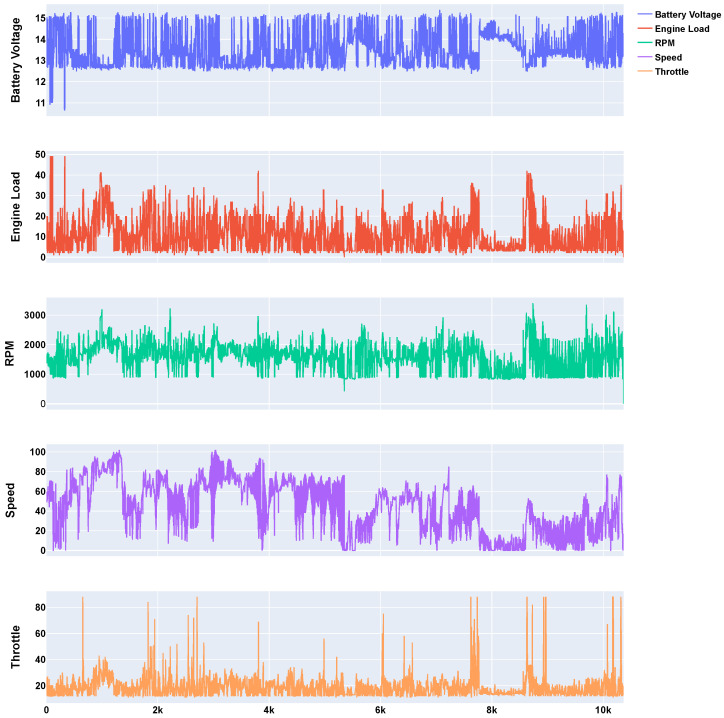
Time series of selected variables: battery voltage, engine load, RPM, speed, and throttle.

**Figure 5 sensors-24-07273-f005:**
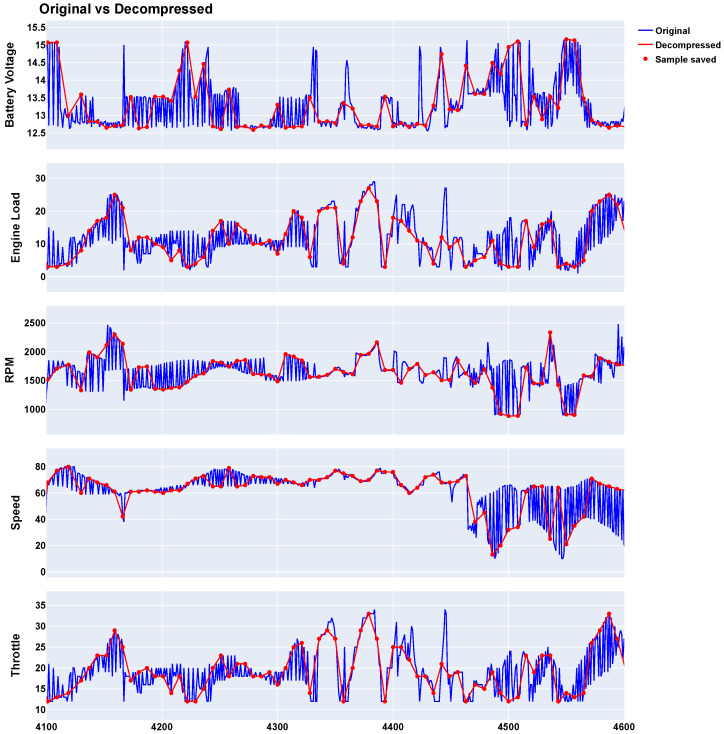
MPTAC compression results (parallel approach).

**Figure 6 sensors-24-07273-f006:**
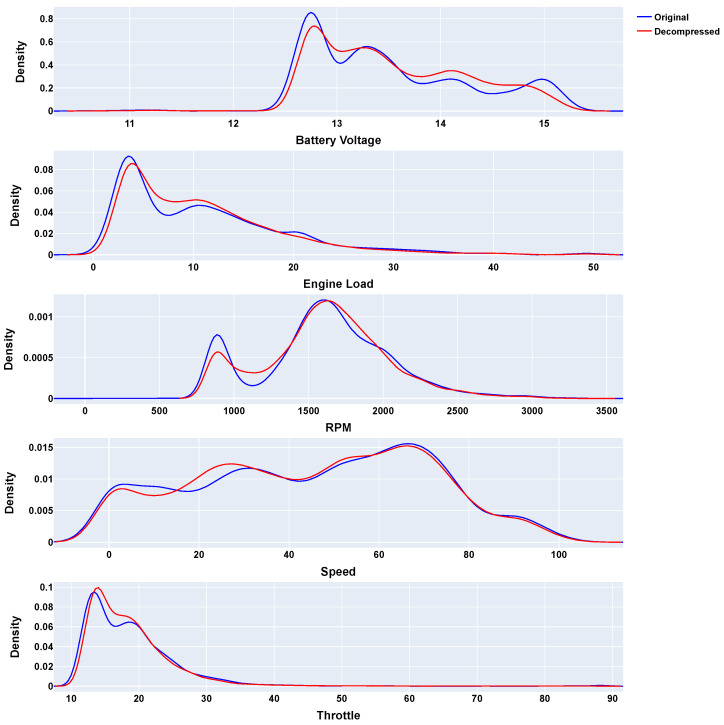
MPTAC compression PDFs (parallel approach).

**Figure 7 sensors-24-07273-f007:**
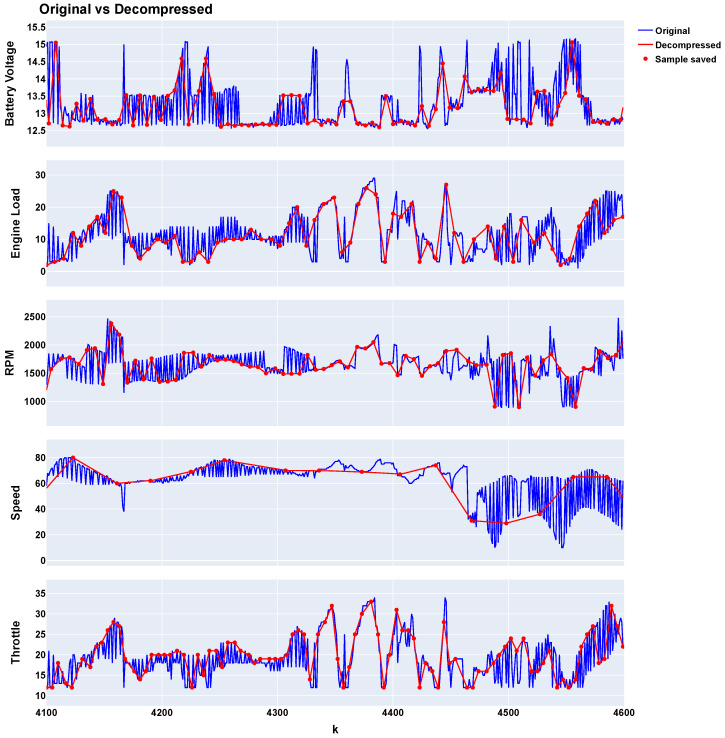
MSTAC compression results (sequential approach).

**Figure 8 sensors-24-07273-f008:**
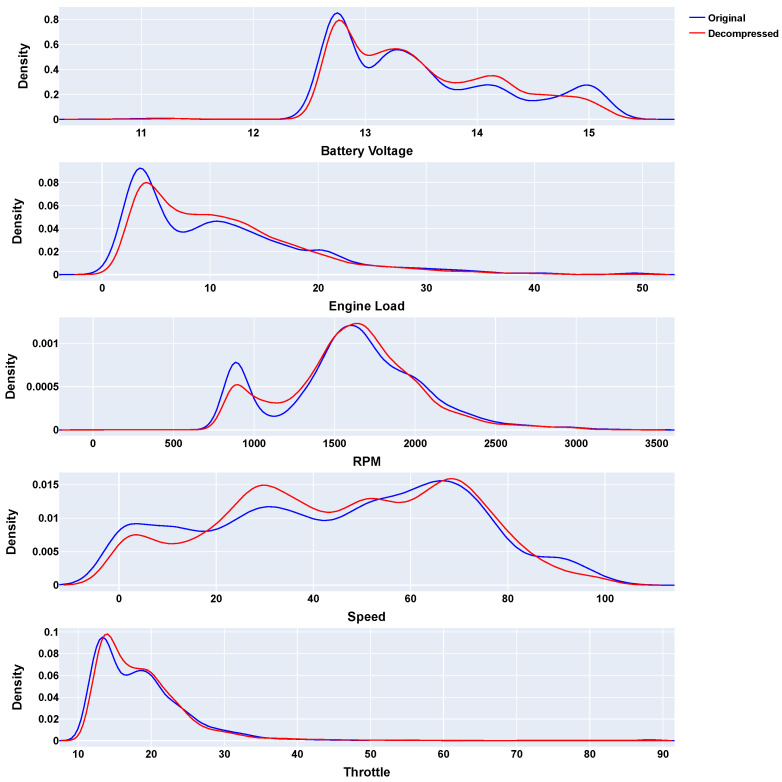
MSTAC compression PDFs (sequential approach).

**Figure 9 sensors-24-07273-f009:**
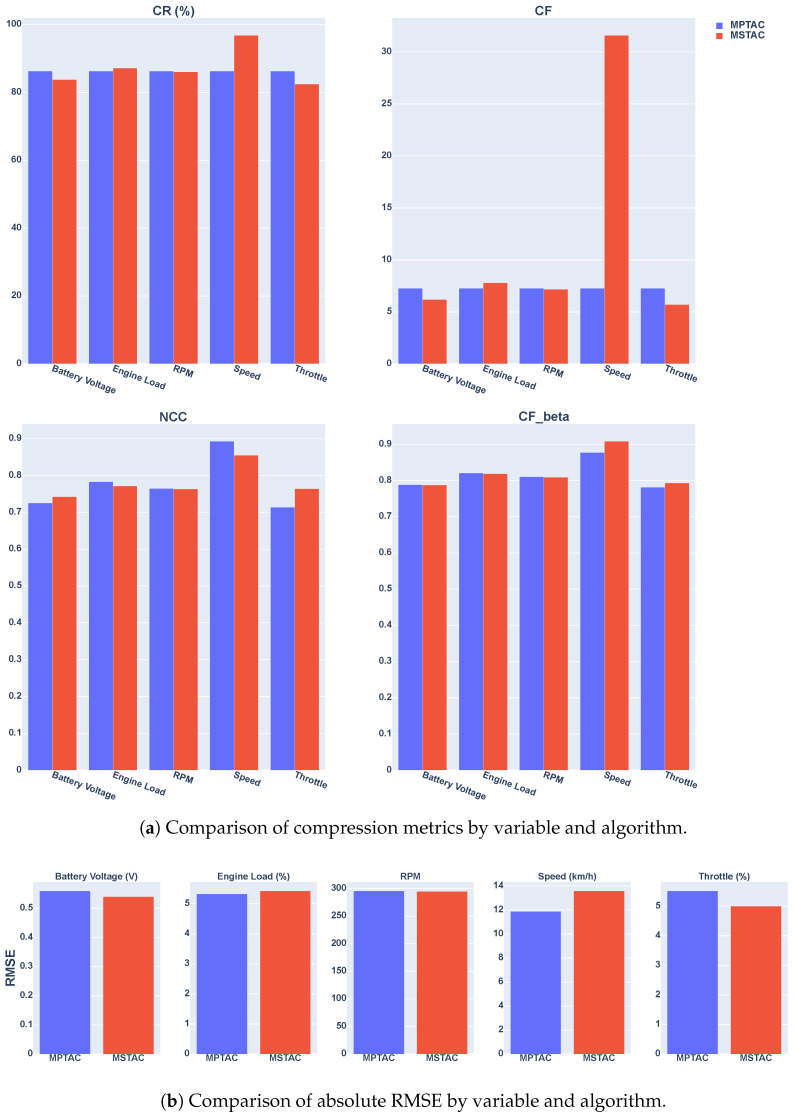
Visual comparison of MPTAC and MSTAC performance metrics, using the same parameters.

**Table 1 sensors-24-07273-t001:** Summary of related works.

Work	Multivariate Input	Float-Point Precision	Online Learning	Machine Learning	1-Stage Compres.
Azar et al. [19]	Yes	Yes	No	No	No
Vox et al. [20]	Yes	No	No	No	No
Ithayarani et al. [21]	Yes	Yes	No	Yes	Yes
Chandak et al. [22]	Yes	Yes	Yes	Yes	No
Barot et al. [23]	Yes	Yes	No	Yes	Yes
Agrawal et al. [24]	Yes	Yes	No	No	No
Meng et al. [25]	No	Yes	No	No	No
Li et al. [26]	No	Yes	No	No	No
Signoretti et al. [12]	No	Yes	Yes	Yes	Yes
Proposed work	Yes	Yes	Yes	Yes	Yes

**Table 2 sensors-24-07273-t002:** Descriptive statistics of the original data for each variable.

	Battery Voltage (V)	Engine Load (%)	RPM	Speed (km/h)	Throttle (%)
**# of samples**	10.365	10.365	10.365	10.365	10.365
**Mean**	13.52	11.07	1608.33	45.42	18.86
**Std**	0.79	8.38	442.92	25.97	7.55
**Min**	10.64	0.00	0.00	0.00	11.00
**Max**	15.39	49.00	3401.00	102.00	88.00

**Table 3 sensors-24-07273-t003:** MPTAC compression metrics (parallel approach).

Variables	*m*	windowLimit	CR (%)	CF	RMSE	NCC	${\mathrm{CF}}_{\beta }$
Battery voltage	0.2	5	86.25	7.27	0.56	0.7256	0.7882
Engine load					5.33	0.7832	0.8210
RPM					295.73	0.7645	0.8106
Speed					11.89	0.8929	0.8774
Throttle					5.52	0.7139	0.7812

**Table 4 sensors-24-07273-t004:** MSTAC compression metrics (sequential approach).

Variables	*m*	windowLimit	CR (%)	CF	RMSE	NCC	${\mathrm{CF}}_{\beta }$
Battery voltage	0.1	4	83.81	6.18	0.54	0.7424	0.7874
Engine load	0.1	5	87.16	7.79	5.43	0.7717	0.8186
RPM	0.1	5	86.08	7.18	294.68	0.7633	0.8091
Speed	0.2	27	96.83	31.60	13.60	0.8552	0.9083
Throttle	0.1	3	82.47	5.70	5.00	0.7638	0.7931

**Table 5 sensors-24-07273-t005:** MPTAC versus MSTAC compression metrics.

Variable		Algorithm	Parameters	CR (%)	CF	RMSE	NCC	${\mathrm{CF}}_{\beta }$
Battery voltage		Parallel	(0.2, 5)	86.25	7.27	0.56	0.7256	0.7882
		Sequential	(0.1, 4)	83.81	6.18	0.54	0.7424	0.7874
ine	Engine load	Parallel	(0.2, 5)	86.25	7.27	5.33	0.7832	0.8210
		Sequential	(0.1, 5)	87.16	7.79	5.43	0.7717	0.8186
ine	RPM	Parallel	(0.2, 5)	86.25	7.27	295.73	0.7645	0.8106
		Sequential	(0.1, 5)	86.08	7.18	294.68	0.7633	0.8091
ine	Speed	Parallel	(0.2, 5)	86.25	7.27	11.89	0.8929	0.8774
		Sequential	(0.2, 27)	96.83	31.60	13.60	0.8552	0.9083
ine	Throttle	Parallel	(0.2, 5)	86.25	7.27	5.52	0.7139	0.7812
		Sequential	(0.1, 3)	82.47	5.70	5.00	0.7638	0.7931

**Table 6 sensors-24-07273-t006:** MPTAC versus MSTAC compression metrics (same parameterization).

Variable		Algorithm	Parameters	CR (%)	CF	RMSE	NCC	${\mathrm{CF}}_{\beta }$
Battery voltage		Parallel	(0.1, 4)	83.53	6.07	0.55	0.7360	0.7825
		Sequential		83.81	6.18	0.54	0.7424	0.7874
ine	Engine load	Parallel	(0.1, 4)	83.53	6.07	5.31	0.7861	0.8100
		Sequential		85.10	6.71	5.49	0.7646	0.8055
ine	RPM	Parallel	(0.1, 4)	83.53	6.07	291.00	0.7694	0.8010
		Sequential		83.72	6.14	289.20	0.7702	0.8023
ine	Speed	Parallel	(0.1, 4)	83.53	6.07	11.72	0.8953	0.8642
		Sequential		84.77	6.57	11.93	0.8913	0.8690
ine	Throttle	Parallel	(0.1, 4)	83.53	6.07	5.23	0.7335	0.7811
		Sequential		85.25	6.78	5.56	0.6835	0.7587

**Table 7 sensors-24-07273-t007:** Advantages and disadvantages of MPTAC and MSTAC algorithms.

Algorithm		Advantages	Disadvantages
**MPTAC**		(1) Superior processing time: 4.17 times faster than MSTAC. (2) Competitive performance in reconstruction metrics, with an average $CF_{\beta }$ of 0.7882, comparable to MSTAC.	(1) Lower maximum compression ratio for certain variables (e.g., speed). (2) May not achieve the same compression ratio for all variables as MSTAC.
ine	**MSTAC**	(1) Achieved higher compression ratios for certain variables, such as speed (96.83%). (2) Good ability to preserve data integrity, with a $CF_{\beta }$ comparable to MPTAC.	(1) Significantly longer processing time compared to MPTAC. (2) Lower compression ratio for some variables when compared to MPTAC.

## Data Availability

Data are contained within the article.

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
