# Peer review of "An Evolving Multivariate Time Series Compression Algorithm for IoT Applications"

_sensors, 2024, doi:10.3390/s24227273_

Round 1
Reviewer 1 Report
Comments and Suggestions for Authors
1. Please highlight the innovations covered in the introduction section.
2. Algorithms can be partially annotated to make it easier for the reader to understand.
3. Check for acronyms in the manuscript; it is recommended that acronym forms be used for words that occur with high frequency; if they occur only 1-2 times, it is recommended that acronyms not be used.
4.Please standardize the citation format of references.
Comments on the Quality of English LanguageThe English language of the introduction can be improved.
Author Response
Reviewer 1
Please highlight the innovations covered in the introduction section.
Thank you for your feedback, we have incorporated the key innovations directly into the introduction section.
Algorithms can be partially annotated to make it easier for the reader to understand.
Thank you for this valuable suggestion. In response, we have annotated the key steps within the algorithms to enhance clarity and make it easier for the reader to follow the main processes. These annotations provide concise explanations for each major calculation and decision point in the algorithm, improving readability and comprehension.
Check for acronyms in the manuscript; it is recommended that acronym forms be used for words that occur with high frequency; if they occur only 1-2 times, it is recommended that acronyms not be used.
We have thoroughly reviewed the manuscript and adjusted the use of acronyms accordingly. All frequently occurring terms have been abbreviated for clarity, while terms that appeared only once or twice have been left in full to ensure readability.
Please standardize the citation format of references.
We have standardized the references throughout the manuscript to ensure consistency in formatting.

Reviewer 2 Report
Comments and Suggestions for Authors
In a previous study titled "An Evolving TinyML Compression Algorithm for IoT Environments Based on Data Eccentricity," a new compression technique called Tiny Anomaly Compressor (TAC) was introduced. TAC is specifically tailored for local and online data compression on IoT-enabled devices, utilizing concepts of typicality and eccentricity without relying on pre-established mathematical models or data distribution assumptions. This approach eliminates the need for complex mathematics or data assumptions, prioritizing typical and unusual data while being efficient in terms of memory and processing power, making it suitable for resource-constrained devices.
This manuscript introduces two multivariate extensions of TAC: MPTAC (Multivariate Parallel TAC) and MSTAC (Multivariate Sequential TAC). These extensions aim to compress time series data in IoT applications, particularly in vehicular contexts. The authors employ the Typicality and Eccentricity Data Analytics (TEDA) framework to enhance data compression without relying on predefined mathematical models. The results indicate considerable improvements in execution time and compression errors, thereby boosting the performance of embedded IoT systems.
The contributions of the paper are important for several reasons: (1) The topic is highly relevant given the exponential growth of IoT devices and the need for efficient data compression, especially in vehicular IoT applications where massive data is generated. (2) The paper clearly outlines the limitations of existing univariate compression techniques like TAC, and justifies the need for multivariate solutions (MPTAC and MSTAC). (3) The proposed algorithms are based on sound principles such as the Typicality and Eccentricity Data Analysis (TEDA) framework, and the paper provides comprehensive mathematical formulations. (4) The use of the OBD-II Freematics ONE+ dataset for vehicle monitoring adds practical relevance to the study.
The research has direct implications for the automotive industry as it improves real-time decision-making and predictive maintenance. This makes the research relevant and likely to attract the attention of the scientific and technological community, contributing to the field of IoT and time series compression. I believe that this paper should be accepted for publication in this journal.
The manuscript is generally well-organized, with a logical flow that guides the reader through the research process. However, it would be beneficial to include a more comprehensive literature review, provide a clearer description of the algorithms, improve the presentation of experimental results, and offer a more in-depth discussion of limitations and future work. Please consider the following recommendations:
· The literature review could benefit from additional detail on multivariate time series compression techniques. While it mentions relevant works, a more comprehensive comparison with the proposed methods would enhance its depth. Furthermore, a more thorough analysis of the limitations of previous research would provide a stronger justification for the new algorithms. I recommend enhancing the analysis of related studies by examining their limitations in scalability, accuracy, and computational efficiency. This will help underscore the significance of MPTAC and MSTAC.
· It may be beneficial to enhance the descriptions of MPTAC and MSTAC by including more detailed pseudocode or step-by-step flowcharts. This could potentially improve the understandability of the algorithms for readers who are not deeply familiar with data compression or anomaly detection. Therefore, it is recommended to consider including detailed pseudocode or flowcharts for both algorithms as this would not only make the work more understandable but also enhance reproducibility.
· The selection of the OBD-II Freematics ONE+ dataset is appropriate. However, it would be beneficial to include more comprehensive information about the dataset's characteristics, including sample size, sensor types, and preprocessing steps. This additional detail would enhance the ability to thoroughly evaluate the generalizability of the findings. I recommend providing a more detailed description of the dataset as it is essential for assessing the potential applicability of the proposed algorithms to other datasets.
· The results show positive developments in execution time and compression errors. Nevertheless, there is an opportunity to further enhance the presentation. The tables and figures would benefit from additional visual elements to facilitate the comparison of performance across various variables. It is recommended to incorporate more visual aids to illustrate performance metrics such as compression ratio, RMSE, and execution time for both algorithms effectively. Including comparative graphs of MPTAC and MSTAC would aid readers in quickly grasping the improvements.
· The manuscript would benefit from a more thorough discussion of the proposed algorithms' limitations. It would be helpful to address how the algorithms perform in highly dynamic environments and how they scale with larger datasets. I recommend considering the addition of a section that discusses the potential limitations of MPTAC and MSTAC, such as scalability or performance issues in different environments. This balanced discussion would enhance the credibility of the research.
· The discussion of future work could benefit from additional details to provide a clearer direction for potential research paths. A more comprehensive vision would help readers understand the broader impact of this research and inspire further investigations.

Author Response
Reviewer 2
The manuscript is generally well-organized, with a logical flow that guides the reader through the research process. However, it would be beneficial to include a more comprehensive literature review, provide a clearer description of the algorithms, improve the presentation of experimental results, and offer a more in-depth discussion of limitations and future work. Please consider the following recommendations:
- The literature review could benefit from additional detail on multivariate time series compression techniques. While it mentions relevant works, a more comprehensive comparison with the proposed methods would enhance its depth. Furthermore, a more thorough analysis of the limitations of previous research would provide a stronger justification for the new algorithms. I recommend enhancing the analysis of related studies by examining their limitations in scalability, accuracy, and computational efficiency. This will help underscore the significance of MPTAC and MSTAC.
To address the need for a more in-depth analysis of multivariate time series compression techniques, we have enhanced the literature review by providing a detailed discussion of each cited work, focusing on their scalability, accuracy, and computational efficiency limitations. These additions underscore the scenarios in which current methods may fall short for IoT and TinyML applications, particularly given data variability characteristics and hardware constraints such as memory and processing power limitations.
As recommended, we included an analysis of the limitations of specific methods like TAC, which, while efficient for univariate compression, poses challenges in multivariate contexts due to its limited flexibility in capturing inter-variable relationships. This discussion clarifies the need for the proposed MPTAC and MSTAC algorithms, which are tailored to overcome these limitations through multivariate support and optimizations specifically designed for the TinyML environment. Additionally, this enhanced review helps position MPTAC and MSTAC as significant advancements by directly addressing the scalability, accuracy, and efficiency constraints identified in related work.
- It may be beneficial to enhance the descriptions of MPTAC and MSTAC by including more detailed pseudocode or step-by-step flowcharts. This could potentially improve the understandability of the algorithms for readers who are not deeply familiar with data compression or anomaly detection. Therefore, it is recommended to consider including detailed pseudocode or flowcharts for both algorithms as this would not only make the work more understandable but also enhance reproducibility.
We appreciate the suggestion to include detailed pseudocode and flowcharts for the MPTAC and MSTAC algorithms to enhance the understanding and reproducibility of the work. We would like to highlight that the manuscript already contains these elements and emphasized them in the text to improve visualization and reader comprehension.
- The selection of the OBD-II Freematics ONE+ dataset is appropriate. However, it would be beneficial to include more comprehensive information about the dataset's characteristics, including sample size, sensor types, and preprocessing steps. This additional detail would enhance the ability to thoroughly evaluate the generalizability of the findings. I recommend providing a more detailed description of the dataset as it is essential for assessing the potential applicability of the proposed algorithms to other datasets.
We would like to clarify that the section "Database Selection" already detailed the dataset's characteristics, including sample size and sensor types. Additionally, we want to emphasize that no preprocessing steps were applied to the dataset, as we used the raw data directly for our analysis.
- The results show positive developments in execution time and compression errors. Nevertheless, there is an opportunity to further enhance the presentation. The tables and figures would benefit from additional visual elements to facilitate the comparison of performance across various variables. It is recommended to incorporate more visual aids to illustrate performance metrics such as compression ratio, RMSE, and execution time for both algorithms effectively. Including comparative graphs of MPTAC and MSTAC would aid readers in quickly grasping the improvements.
Thank you for your valuable suggestion. We have added visual representations of the performance metrics to enhance the interpretability of the results.
- The manuscript would benefit from a more thorough discussion of the proposed algorithms' limitations. It would be helpful to address how the algorithms perform in highly dynamic environments and how they scale with larger datasets. I recommend considering the addition of a section that discusses the potential limitations of MPTAC and MSTAC, such as scalability or performance issues in different environments. This balanced discussion would enhance the credibility of the research.
Thank you for your feedback regarding the proposed algorithms' limitations. In response, we have added a new subsection in the Results section that discusses the limitations of both the MPTAC and MSTAC algorithms. This subsection addresses how the algorithms perform in highly dynamic environments and outlines scalability concerns when dealing with larger datasets.
- The discussion of future work could benefit from additional details to provide a clearer direction for potential research paths. A more comprehensive vision would help readers understand the broader impact of this research and inspire further investigations.
We have expanded this section in the manuscript to provide a clearer direction for potential research paths. The revised paragraph now emphasizes exploring the application of MPTAC and MSTAC across various datasets and contexts to assess their versatility and robustness. Additionally, it includes an investigation into the adaptability of these algorithms to different window sizes and parameters, as well as analyses of their application in real-time environments and on hardware platforms such as the Freematics ONE+. These enhancements highlight the broader impact of this research and aim to inspire further investigations.

Reviewer 3 Report
Comments and Suggestions for Authors
The authors presented two multivariate compression approaches based on the Typicality and Eccentricity Data Analytics (TEDA) framework and tried to improve data compression for resource-constrained IoT devices. The methods are tested on automotive data and demonstrate improvements in execution time and compression performance.
The paper addresses an important topic in IoT and TinyML, but it requires major revisions. The novelty of the work needs to be more clearly differentiated from existing methods. Moreover the theoretical foundations need further clarification. Additionally, the results section lacks sufficient depth in discussing trade-offs between compression efficiency and data accuracy. Some improvements in presentation, clarity and theoretical explanation required before publication.
Generally, some technical terms are not well-defined and there are some English phrasing issues that can be resolved by some general editing. More specific comments are listed below.
Section 1: Introduction
1. Please clarify how the proposed methods are different from other existing methods in TinyML and multivariate data compression.
2. Explain more on the specific IoT scenarios that the proposed approach is more beneficial not only for vehicle monitoring.
3. That would be great if authors can provide more justification for using TinyML compare to other ml methods in terms of energy consumption.
4. State the main research problem more clearly in the introduction section in a way that readers who don’t wish to read the entire paper can still grasp the key points of the study.
Section 2: Related Works
5. Identify the main gaps in the existing methods where the proposed approach can address more specifically in TinyML multivariate data compression.
Section 3: Theoretical Background
6. The explanation of eccentricity in TEDA was unclear to me. Some explanation that how eccentricity applies to anomaly detection can help reader to follow better.
7. Moreover clarify the underlying assumptions that TEDA needs and justify the choice of thresholds in anomaly detection.
Section 4: Proposed Algorithm
8. I believe that the computational complexity of MPTAC and MSTAC should be discussed with a clear comparison of their scalability to larger datasets.
9. Include an analysis of how sensitive the algorithms are to the choice of parameters (for example window size, m-threshold).
10. Please include how the proposed algorithms handle strong correlations between variables?
11. Handling of edge cases (for example data with no anomalies, highly erratic time series) is not adequately addressed in the algorithm description.
Section 5: Case Study
12. The case study uses automotive data. How are these results can be generalized to other IoT applications?
13. Include some details on the decompression errors that are observed in the case study and the impact on the applications.
14. Explain if any data preprocessing (for example, normalization or filtering) was applied before applying the proposed algorithms and how that impacts performance.
Section 6: Results
15. I believe the results can be more interpretable by some visual summaries of performance metrics if it's possible.
16. Discuss on the trade offs between compression ratio and reconstruction accuracy and how this impacts IoT applications.
17. More explanation on how parameters were optimized is beneficial. Also explain whether this optimization is feasible in real-world settings. Is it global?
18. Address how outliers or highly anomalous data points affect the compression and decompression processes.
The paper addresses an important topic in IoT and TinyML, but it requires major revisions.
Author Response
Reviewer 3
Section 1: Introduction
- Please clarify how the proposed methods are different from other existing methods in TinyML and multivariate data compression.
We clarified the distinctions between the proposed methods (MPTAC and MSTAC) by explaining that these algorithms were developed to address the limitations of univariate techniques, such as the original TAC when dealing with multivariate data. The introduction section was expanded to explain how MPTAC and MSTAC address the interaction and correlation between variables, something that univariate techniques cannot accurately capture. We also included a comparison with conventional ML techniques, highlighting the advantages of TinyML in resource-constrained IoT devices.
- Explain more on the specific IoT scenarios that the proposed approach is more beneficial not only for vehicle monitoring.
The introduction was revised to highlight the applicability of our methodology in other IoT scenarios, including environmental monitoring, industrial automation, and smart energy grids. These cases share similar challenges, such as the large volume of multivariate data and the need for efficient compression on resource-constrained devices.
- That would be great if authors can provide more justification for using TinyML compare to other ml methods in terms of energy consumption.
In the introduction, we have included a more detailed justification for the advantages of using TinyML in IoT devices, highlighting its lower energy consumption than conventional ML techniques. TinyML's energy efficiency and the possibility of local processing reduce the need for continuous data transmission and save energy in IoT scenarios, making the approach especially advantageous in applications with limited battery and connectivity.
- State the main research problem more clearly in the introduction section in a way that readers who don’t wish to read the entire paper can still grasp the key points of the study.
Following the suggestion, we have reinforced the introduction with a more precise explanation of the central problem. The research problem has now been made explicit by describing how univariate techniques, such as TAC, are limited when dealing with data from multiple interconnected variables, especially in IoT contexts that require compression and real-time anomaly detection. The text now clearly presents the central contributions so that readers can understand the objectives and motivations of the study without needing to read the full paper.
Section 2: Related Works
- Identify the main gaps in the existing methods where the proposed approach can address more specifically in TinyML multivariate data compression.
To address the identified gaps in existing methods for multivariate data compression in TinyML applications, we have expanded the literature review to emphasize specific limitations, particularly scalability, computational efficiency, and adaptability to multivariate data. The reviewed methods often demonstrate challenges in accurately capturing inter-signal relationships, handling non-stationary data, and efficiently compressing high-dimensional datasets under constrained memory and processing conditions. This gap underscores the necessity for a solution like MPTAC and MSTAC, which enhance the traditional TAC algorithm by incorporating multivariate support and optimizations for TinyML’s resource limitations. Our methods are designed to maintain compression efficiency while minimizing computational demands, making them suitable for low-power IoT devices where existing methods may fall short.
Section 3: Theoretical Background
- The explanation of eccentricity in TEDA was unclear to me. Some explanation that how eccentricity applies to anomaly detection can help reader to follow better.
Thank you for your insightful feedback. In response, we have expanded Section 3 to clarify how eccentricity and typicality function within TEDA and their roles in anomaly detection. We added explanations to show how eccentricity measures deviations in data points, while typicality reflects the representativeness of each point within the dataset. These additions aim to enhance the reader's understanding of TEDA’s theoretical foundation and its application to anomaly detection.
- Moreover clarify the underlying assumptions that TEDA needs and justify the choice of thresholds in anomaly detection.
Thank you for your valuable suggestion. In response, we have expanded Section 3 to clarify the underlying assumptions TEDA requires and the rationale behind our choice of thresholds for anomaly detection. TEDA operates without traditional assumptions such as independence, large sample sizes, or fixed data distributions, making it suitable for real-time and dynamic data streams. The non-parametric nature of TEDA allows it to detect anomalies by examining data proximity and mutual distribution rather than relying on a predefined statistical model. The threshold is set to enable sensitivity to deviations, providing flexibility to adapt to data variability and effectively balance detection accuracy.
Section 4: Proposed Algorithm
- I believe that the computational complexity of MPTAC and MSTAC should be discussed with a clear comparison of their scalability to larger datasets.
We have addressed the computational complexity and scalability of MPTAC and MSTAC in the added subsection, “Algorithm Characteristics of MPTAC and MSTAC.” Both algorithms have a computational complexity of O(n), where n is the number of monitored variables. This characteristic ensures that their performance scales linearly with data dimensionality. Furthermore, MPTAC’s design utilizes parallel processing, which enhances its practical efficiency, particularly in high-dimensional applications typical of IoT and multivariate data systems. This is explicitly discussed in the section, ensuring clarity on how both algorithms manage scalability to larger datasets.
- Include an analysis of how sensitive the algorithms are to the choice of parameters (for example window size, m-threshold).
The subsection, “Algorithm Characteristics of MPTAC and MSTAC” --- also provides a thorough analysis of parameter sensitivity, focusing on the anomaly window size and m-threshold. It describes how the anomaly window size determines the responsiveness of the algorithms to detected anomalies, where smaller sizes allow for quicker anomaly capture but may increase false positives in volatile data. Similarly, the m-threshold is detailed as a critical factor in defining the sensitivity to variations, enabling users to adjust detection levels based on the characteristics of their datasets.
- Please include how the proposed algorithms handle strong correlations between variables?
The handling of solid correlations between variables is explicitly addressed in the subsection, “Algorithm Characteristics of MPTAC and MSTAC”. MPTAC calculates a single eccentricity score for the multivariate sample, effectively capturing collective patterns among correlated variables. This is advantageous for datasets where relationships between variables are significant. Conversely, MSTAC treats each variable independently, which enhances its ability to detect individual deviations. Additionally, the discussion includes the strategy of selecting representative variables in resource-constrained scenarios, like IoT, to optimize processing by focusing on the most informative variables while avoiding redundancy.
- Handling of edge cases (for example data with no anomalies, highly erratic time series) is not adequately addressed in the algorithm description.
The new subsection, “Algorithm Characteristics of MPTAC and MSTAC” discusses how MPTAC and MSTAC manage edge cases, such as very smooth and highly erratic data streams. The algorithms can increase sensitivity to subtle anomalies by lowering the m-threshold and adjusting the anomaly window size for smooth data. This ensures effective anomaly detection while maintaining data integrity during compression. In the case of highly erratic time series, raising these parameters helps prevent excessive data retention, allowing the algorithms to focus on significant deviations.
Section 5: Case Study
- The case study uses automotive data. How are these results can be generalized to other IoT applications?
We included a paragraph detailing how the results can be generalized to other IoT applications. This section highlights that the findings are not limited to the automotive context but also apply to various scenarios, such as wearable devices and smart city infrastructures. By emphasizing the importance of processing speed and compression ratios across different applications, we aimed to demonstrate the broader relevance of the study's conclusions in the IoT landscape.
- Include some details on the decompression errors that are observed in the case study and the impact on the applications.
Based on your suggestion, we have expanded the Compression error subsection to include a discussion of how decompression errors influence the application of reconstructed data in different contexts, especially in systems that require high data fidelity, such as real-time monitoring and predictive maintenance.
- Explain if any data preprocessing (for example, normalization or filtering) was applied before applying the proposed algorithms and how that impacts performance.
We would like to clarify that the characteristics of the dataset, including sample size and sensor types, are detailed in the "Database Selection" section of the manuscript. Importantly, we did not apply any preprocessing steps, such as normalization or filtering, and instead used the raw data directly for our analysis. This decision was driven by the algorithms' design for real-time operation, reflecting their intended application in scenarios where immediate processing of raw sensor data.
Section 6: Results
- I believe the results can be more interpretable by some visual summaries of performance metrics if it's possible.
Following this recommendation, we have included a visual representation of the performance metrics of the MPTAC and MSTAC algorithms in Figure 9. This figure now organizes the visualizations into subfigures to facilitate direct comparison between variables and algorithms.
- Discuss on the trade offs between compression ratio and reconstruction accuracy and how this impacts IoT applications.
We have incorporated an analysis of the trade-offs between compression and accuracy in the results discussion section. The text addresses scenarios where a high compression ratio is beneficial, as well as situations where data reconstruction accuracy becomes a priority, considering the IoT application context.
- More explanation on how parameters were optimized is beneficial. Also explain whether this optimization is feasible in real-world settings. Is it global?
We have added an explanation of the parameter optimization process, with an emphasis on the adaptability of tuning for different domains. We also addressed the feasibility of this optimization in real-world settings, noting that the process is not global but rather specific to the characteristics of the data and application requirements.
- Address how outliers or highly anomalous data points affect the compression and decompression processes.
Thank you for your feedback regarding the proposed algorithms' limitations. In response, we have added a new subsection in the Results section that discusses the limitations of both the MPTAC and MSTAC algorithms.
